# FAM150A and FAM150B are activating ligands for anaplastic lymphoma kinase

Jikui Guan[1†], Ganesh Umapathy[1†], Yasuo Yamazaki[1], Georg Wolfstetter[1], Patricia Mendoza[1], Kathrin Pfeifer[1], Ateequrrahman Mohammed[1], Fredrik Hugosson[1], Hongbing Zhang[2], Amy W Hsu[2], Robert Halenbeck[2], Bengt Hallberg[1], Ruth H Palmer[1]*

[1]Department of Medical Biochemistry and Cell Biology, Instititute of Biomedicine, Sahlgrenska Academy, University of Gothenburg, Gothenburg, Sweden; [2]Five Prime Therapeutics Inc., South San Francisco, United States

**Abstract** Aberrant activation of anaplastic lymphoma kinase (ALK) has been described in a range of human cancers, including non-small cell lung cancer and neuroblastoma (Hallberg and Palmer, 2013). Vertebrate ALK has been considered to be an orphan receptor and the identity of the ALK ligand(s) is a critical issue. Here we show that FAM150A and FAM150B are potent ligands for human ALK that bind to the extracellular domain of ALK and in addition to activation of wild-type ALK are able to drive 'superactivation' of activated ALK mutants from neuroblastoma. In conclusion, our data show that ALK is robustly activated by the FAM150A/B ligands and provide an opportunity to develop ALK-targeted therapies in situations where ALK is overexpressed/activated or mutated in the context of the full length receptor.

## Introduction

*For correspondence: ruth.
palmer@gu.se

Competing interests: The author
declares that no competing
interests exist.

Reviewing editor: Roger Davis,
Howard Hughes Medical Institute
& University of Massachusetts
Medical School, United States

Activation of anaplastic lymphoma kinase (ALK) is commonly due to fusion of the ALK kinase domain with a dimerization partner that drives activation, however, ALK activation also occurs in the context of the full length receptor, for example, as activating point mutations in neuroblastoma (*Maris et al., 2007*; *Carén et al., 2008*; *Chen et al., 2008*; *George et al., 2008*; *Janoueix-Lerosey et al., 2008*; *Mosseé et al., 2008*; *Hallberg and Palmer, 2013*). In many additional tumor types ALK overexpression and activation has been described, and it is unclear whether this is dependent on the activity of a ligand (*Hallberg and Palmer 2013*). The ALK receptors in both *Drosophila* and *Caenorhabditis elegans* have well defined ligands – Jeb (*Englund et al., 2003*; *Lee et al., 2003*; *Stute et al., 2004*) and HEN-1 (*Ishihara et al., 2002*), respectively. In contrast, vertebrate ALK has long been considered as an orphan receptor.

The human *ALK* locus encodes a classical receptor tyrosine kinase (RTK) comprising a unique extracellular ligand-binding domain, a transmembrane domain and an intracellular tyrosine kinase domain (*Hallberg and Palmer, 2013*). The extracellular portion of ALK which contains two MAM domains (named after meprin, A-5 protein and receptor protein tyrosine phosphatase μ), a glycine-rich region (GR) and a LDLa domain, is unique among the RTKs. ALK, and the related leukocyte tyrosine kinase (LTK) RTK, share kinase domain similarities as well as a GR in the membrane proximal portion of their extracellular domains (ECDs) (*Iwahara et al., 1997*; *Morris et al., 1997*). Recent screening of the extracellular proteome identified two novel secreted proteins as ligands for LTK – family with sequence similarity 150A (FAM150A) and family with sequence similarity 150B (FAM150B). Both bind to the ECD of the receptor leading to activation of downstream signaling in cell culture models (*Zhang et al., 2014*). FAM150A and FAM150B are unique, displaying homology only with one another but not with any other proteins in mammals (*Zhang et al., 2014*).

**eLife digest** Cells have receptor proteins on their surface that enable them to detect changes in their environment and communicate with other cells. Signal molecules bind to a segment of the receptor called the extracellular domain that faces out from the cell. This can result in the activation of another domain in the receptor that is just inside the cell, which, in turn, activates signaling pathways that relay the information around the cell.

However, these communication systems are often disrupted in cancer cells. This helps the cells to override the strict growth controls imposed upon them by other (healthy) cells in the body. The gene that encodes a receptor protein called Anaplastic Lymphoma Kinase (or ALK for short) is often mutated in some types of human cancer so that the protein is always active. However, we still do not know what signal molecules bind to the ALK protein to activate it in normal cells.

Guan, Umapathy et al. used a variety of cell biology and biochemical techniques to study the role of ALK. The experiments show that when either of two proteins called FAM150A and FAM150B are produced in rat nerve cells alongside ALK, the nerve cells rapidly respond and form outgrowths. Experiments using cancer cells derived from human nerve cells also yielded similar results. Guan, Umapathy et al. found that the extracellular domain of ALK can physically interact with FAM150A and FAM150B.

The eyes of fruit flies that had been genetically modified to produce the human ALK protein alongside either FAM150A or FAM150B grew more than normal, giving the eyes an abnormal "rough" appearance. Further experiments showed that FAM150A and FAM150B are also able to increase the level of activation of an ALK mutant protein that is already active. Therefore, in future, the development of drugs that stop FAM150A and FAM150B from binding to ALK may be useful for treating cancers that are driven by high levels of ALK activity. Many challenging questions lie ahead to better understand how FAM150A and FAM150B interact with ALK.

Furthermore, we found the reported strong expression of FAM150B in the human adrenal gland (*Zhang et al., 2014*) intriguing, given the role of ALK in neuroblastoma.

Here we report the identification of FAM150A and FAM150B as potent ligands for human ALK. We investigated ALK activation by FAM150A and FAM150B proteins in PC12 cell neurite outgrowth assays where we observed a strong activation of ALK signaling. Conditioned medium containing either FAM150A or FAM150B was able to activate endogenous ALK signaling in neuroblastoma cells. We also employed the model organism *Drosophila melanogaster* as a readout for activation of ALK by FAM150A and FAM150B, showing that FAM150 proteins are able to robustly drive human ALK activation when ectopically coexpressed in the fly. FAM150A and FAM150B bind to the ECD of ALK and, in addition to activation of wild-type ALK, are able to drive 'superactivation' of activated ALK mutants from neuroblastoma. The GR of the ALK receptor ECD is important for FAM150 activation, and monoclonal antibodies (mAb) recognizing the GR of ALK are able to inhibit activation of ALK by FAM150A. In conclusion, our data show that ALK is robustly activated by FAM150A/B finally providing an answer to the identity of the elusive ligands for this RTK.

## Results and discussion

ALK and the related LTK share similarity in their membrane proximal ECD in the form of a glycine-rich domain that is ~250 amino acids in length (*Figure 1A*, GR depicted in grey). This domain contains multiple runs of up to eight glycine residues, and is unique to ALK and LTK within the human genome. The importance of the GR in ALK has been highlighted in *Drosophila* studies, where four independent point mutations leading to exchange of single glycine residues result in complete loss of function *in vivo* (*Englund et al., 2003*) (*Figure 1—figure supplement 1*). The similarity between ALK and LTK within the GR is ~70%, with amino acid identity of 55%, containing a total of 51 conserved glycine residues (*Figure 1B*). Given this similarity, and the important role of the glycine-rich domain for function in *Drosophila*, we hypothesized that FAM150A and FAM150B, which were recently reported as ligands for LTK (*Zhang et al., 2014*) may act as ligands for ALK.

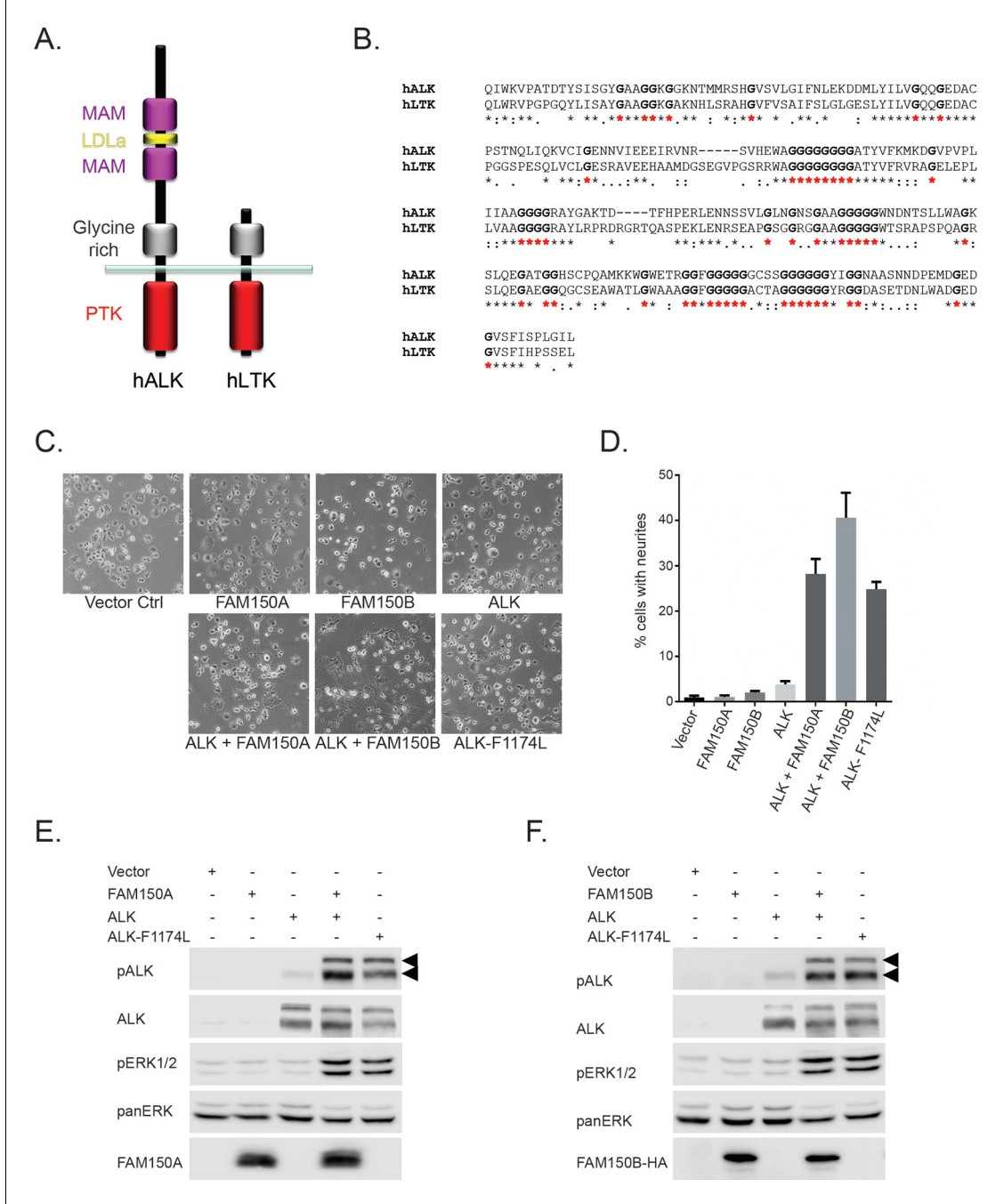

**Figure 1.** FAM150A and FAM150B activate ALK. (**A**) Schematic overview of human anaplastic lymphoma kinase (ALK) and leukocyte tyrosine kinase (LTK) protein domain structures. ALK and LTK share a membrane proximal extracellular glycine-rich region (GR, grey), transmembrane and an intracellular tyrosine kinase domain (red). In addition, the extracellular region of ALK contains two MAM domains (purple) and an LDLa-motif (yellow). (**B**) Alignments of the GR of ALK and LTK, conserved runs of glycine residues are highlighted in bold with red asterisks. (**C**) Neurite outgrowth in PC12 cells expressing either vector control, FAM150A, FAM150B, ALK, FAM150A and ALK, FAM150B and ALK or ALK-F1174L quantified in (**D**). Experiments were performed in triplicate and each sample within an experiment was performed in duplicate (error bars indicate SD). (**E**) Whole cell lysates from PC12 cells expressing either vector control, FAM150A, ALK, FAM150A and ALK or ALK-F1174L were analyzed by immunoblot analysis of ALK, pALK-Y1604 (arrowheads), FAM150A and ERK1/2. Pan-ERK was employed for equal loading. (**F**) Whole cell lysates from PC12 cells expressing either vector control, FAM150B, ALK, FAM150B

*Figure 1. continued on next page*

*Figure 1. Continued*

and ALK or ALK-F1174L were analyzed by immunoblot analysis of ALK, pALK-Y1604 (arrowheads), HA (FAM150B) and pERK1/2. Pan-ERK was employed for equal loading.

The following figure supplements are available for Figure 1:

**Figure supplement 1.** Glycine residues in the glycine-rich region of *Drosophila* ALK are critical for function.

**Figure supplement 2.** Conservation of phosphoepitopes in the intracellelular domains of ALK and LTK.

In order to initially test whether FAM150A or FAM150B could activate ALK we assayed neurite outgrowth activity in PC12 cells. Expression of either FAM150A, or FAM150B or ALK alone, did not lead to activation. This was in contrast to the strong activation seen with the positive control ALK-F1174L which is a well characterized constitutively active ALK neuroblastoma mutation. However, coexpression with either FAM150A or FAM150B led to robust activation of ALK neurite outgrowth activity (*Figure 1C*, quantified in *1D*). This ALK activation was visualized by phosphorylation of Y1604 in the tail of the ALK intracellular domain (*Figure 1E, F*; *Figure 1—figure supplement 2*) and was further associated with stimulation of downstream signaling such as the phosphorylation of the downstream target ERK1/2 (*Figure 1E, F*). Thus in PC12 cells FAM150A and FAM150B act to stimulate signaling via the wild-type ALK receptor, in a manner that leads to phosphorylation of the ALK receptor.

We next tested the ability of exogenously produced FAM150A or FAM150B to activate ALK, employing medium conditioned with FAM150A or FAM150B to activate PC12 cells expressing ALK. Addition of conditioned medium from cells expressing FAM150A or FAM150B led to strong activation of ALK in receiving cells that expressed ALK, as measured by neurite outgrowth and activation of downstream signaling (*Figure 2A, B*). We then examined whether ALK signaling activity induced by either FAM150A or FAM150B was sensitive to ALK inhibition, employing the ALK inhibitor crizotinib (*Zou et al., 2007*). We observed that ALK activation by FAM150A or FAM150B was inhibited by addition of 250 nM crizotinib (*Figure 2A, B*).

To address whether endogenous levels of ALK responded to medium conditioned with either FAM150A or FAM150B, we employed the IMR-32 neuroblastoma cell line that expresses wild-type ALK. Once again, we observed robust activation of ALK by the addition of either FAM150A-containing or FAM150B-containing conditioned medium, employing pALK-Y1604 and pERK1/2 as readout (*Figure 2C*). The activation of ALK and downstream signaling events in IMR-32 cells were effectively blocked by the addition of 250 nM crizotinib (*Figure 2C*). The activation of exogenous ALK in PC12 cells or endogenous ALK in IMR-32 cells was reproduced with the addition of recombinant FAM150A purified from Sf21 cells (*Figure 2D*). Unfortunately we could not produce sufficient amounts of recombinant FAM150B protein, precluding a similar analysis with FAM150B.

More careful examination of ALK activation by FAM150A or FAM150B containing conditioned media in IMR-32 cells revealed activation of signaling at 2–5 min, peaking around 20 min before decreasing at later time points. This response was observed at the level of ALK receptor phosphorylation (pALK-Y1604), and with downstream pERK5 (*Umapathy et al., 2014*) and pERK1/2 (*Figure 2E, F*). Here we employed antibodies recognizing pALK-Y1604, allowing differentiation of ALK activation in IMR-32 cells from that of LTK (*Figure 1—figure supplement 2*). Thus we observed that exogenously produced FAM150A and FAM150B proteins stimulate ALK signaling in both PC12 cells and IMR-32 cells. We also examined potential modulation of ALK activation by FAM150A/B upon addition of heparin, which has recently been reported to activate ALK (*Murray et al., 2015*). While we were able to see strong binding of purified FAM150A protein as well as FAM150A-HA and FAM150B-HA from conditioned medium to heparin-agarose, we did not observe any additional activation of ALK by FAM150 proteins in the presence of heparin (*Figure 2—figure supplement 1; Figure 2—figure supplement 2*).

We next analyzed the ability of FAM150A or FAM150B to activate human ALK in a *Drosophila* model, which offers a clear readout. Neither the *Drosophila* Alk ligand Jeb (*Englund et al., 2003*; *Lee et al., 2003*; *Stute et al., 2004*) nor previously proposed vertebrate ligands, that is, human midkine (MDK) and pleiotrophin (PTN) are able to activate either mouse or human ALK (*Yang et al.,*

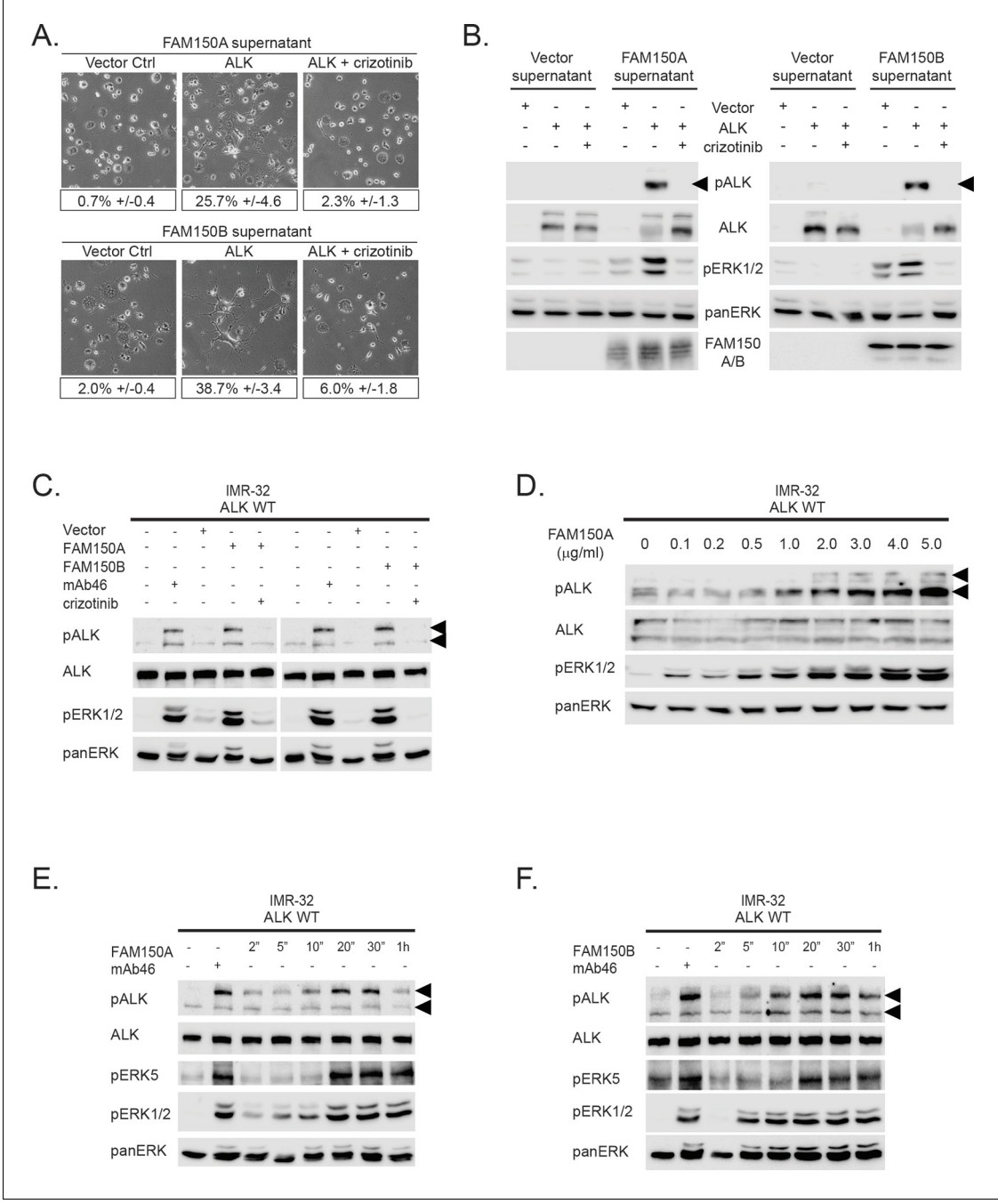

**Figure 2.** Conditioned medium containing either FAM150A or FAM150B activates endogenous ALK. (**A**) Neurite outgrowth in PC12 cells expressing either vector control or anaplastic lymphoma kinase (ALK) were cultured in medium from Human Embryonic Kidney (HEK) 293 cells transfected with either FAM150A or FAM150B, quantified below. Experiments were performed in triplicate and each sample within an experiment was performed in duplicate. Values represent mean ± SD from at least three independent experiments. (**B**) Whole cell lysates from PC12 cells expressing either vector control or ALK stimulated with medium from HEK293 cells transfected with vector control, FAM150A or FAM150B were analyzed by immunoblot. Analysis was carried out in the presence or absence of 250 nM crizotinib. Detection of ALK activation was visualized with pALK-Y1604 (arrowheads), ALK and pERK1/2 in whole cell lysates. The presence of FAM150A in supernatants was confirmed with anti-FAM150A antibodies, while the presence of FAM150B-HA was confirmed with anti-HA antibodies. Pan-ERK was employed for equal loading. (**C**) IMR32 cells harboring a wild-type ALK receptor were stimulated for 20 min with medium from HEK293 cells transfected with either vector control, FAM150A or FAM150B prior to analysis by immunoblot.

*Figure 2. continued on next page*

*Figure 2. Continued*

Analysis was carried out in the presence or absence of 250 nM crizotinib. Stimulation with the ALK activating antibody mAb46 was employed as positive control. Detection of ALK activation was visualized with ALK, pALK-Y1604 (arrowheads) and pERK1/2. Pan-ERK was employed for equal loading. (D) IMR32 cells harboring a wild-type ALK receptor stimulated with increased amounts of recombinant His-tagged FAM150A purified from Sf21 cells. Detection of ALK activation was visualized with ALK, pALK-Y1278 (arrowheads) and pERK1/2. Pan-ERK was employed for equal loading. (E) Time course of IMR32 cells stimulated with FAM150A conditioned medium. Stimulation with ALK activating antibody mAb46 was employed as positive control. Detection of ALK activation was visualized with ALK, pALK-Y1604 (arrowheads), pERK5 and pERK1/2. Pan-ERK was employed for equal loading. (F) Time course of IMR32 cells harboring a wild type ALK receptor stimulated with FAM150B conditioned medium. Stimulation with ALK activating antibody mAb46 was employed as positive control. Detection of ALK activation was visualized with ALK, pALK-Y1604 (arrowheads), pERK5 and pERK1/2. Pan-ERK was employed for equal loading.

The following figure supplements are available for Figure 2:

**Figure supplement 1.** FAM150 proteins interact with heparin.

**Figure supplement 2.** Investigation of the effect of Heparin and FAM150A on ALK activation.

---

*2007*; *Hugosson et al., 2014*). Expression of either FAM150A or FAM150B in the developing eye, using the *GMR-Gal4* driver, resulted in normal eye morphology (*Figure 3A*). In contrast, expression of constitutively active ALK-F1174S described in neuroblastoma patients results in a rough eye morphology (*Figure 3A*) (*Martinsson et al., 2011*), while no eye phenotype was observed upon the expression of wild-type human ALK alone (*Figure 3A*) (*Martinsson et al., 2011*, *Schonherr, Ruuth et al., 2011*, *Schonherr, Ruuth et al., 2011*, *Chand et al., 2013*; *Hugosson et al., 2014*). This can be compared with coexpression of either FAM150A or FAM150B together with human ALK which led to a rough eye phenotype, proving that both FAM150A and FAM150B were able to activate human ALK in this *in vivo* system (*Figure 3A, B*).

We next examined binding of the human ALK ECD with purified FAM150A by ELISA and Biacore surface plasmon resonance (SPR) analysis. Both ELISA and Biacore analysis showed that FAM150A binds specifically to the ALK ECD (*Figure 4A*; *Figure 4—figure supplement 1*). Purified human FAM150A (*Zhang et al., 2014*) binds to ALK-ECD-Fc with a dissociation constant of ~20 nM (*Figure 4A*). We also examined the ability of either FAM150A or FAM150B to bind human ALK by immunoprecipitation and immunofluorescence analysis (*Figure 4B–D*). Here we observed that both FAM150A and FAM150B could be independently immunoprecipitated with the ALK receptor, and that ALK could be immunoprecipitated with either FAM150A or FAM150B in the reciprocal pull-down experiments (*Figure 4B, C*; *Figure 4—figure supplement 2*). Since FAM150A/B also bind LTK, we asked whether ALK and LTK interacted in the presence of FAM150A. Indeed, both ALK and FAM150A were found to immunoprecipitate with LTK (*Figure 4—figure supplement 3*). Immunofluorescence analysis of cells expressing human ALK confirmed these findings, with clear association of both HA-tagged FAM150A and HA-tagged FAM150B with ALK expressing cells observed (*Figure 4D*). In these experiments we observed uptake of FAM150A and FAM150B into ALK-positive intracellular vesicles, suggesting that binding to ALK on the cell surface leads to uptake and internalization of the ALK-FAM150 complex (*Figure 4D*).

We further examined the role of the GR of ALK in FAM150A/B binding. Deletion of the GR in the ALK ECD did not affect FAM150A binding but led to loss of FAM150B binding (*Figure 4——figure supplement 4*). Given the importance of the GR of ALK in *Drosophila* (*Englund et al., 2003*) (*Figure 1——figure supplement 1*), we mutated these conserved residues in human ALK (G740D, G823D, G893E and G934D) and tested their effect on the FAM150-ALK interaction. Despite being well expressed, all four glycine mutations in the GR severely impaired binding of FAM150B to ALK. Consistent with our earlier results we were unable to see any effect of mutation of individual glycine on the FAM150A-ALK interaction (*Figure 4—figure supplement 5*).

In previous work, we and others have generated antibodies to the ECD of ALK (*Moog-Lutz et al., 2005*, *Witek et al., 2015*). We investigated whether any of these antibodies recognized the GR in the ALK ECD, and were able to identify three interesting candidates (anti-ALK mAb13, mAb48 and

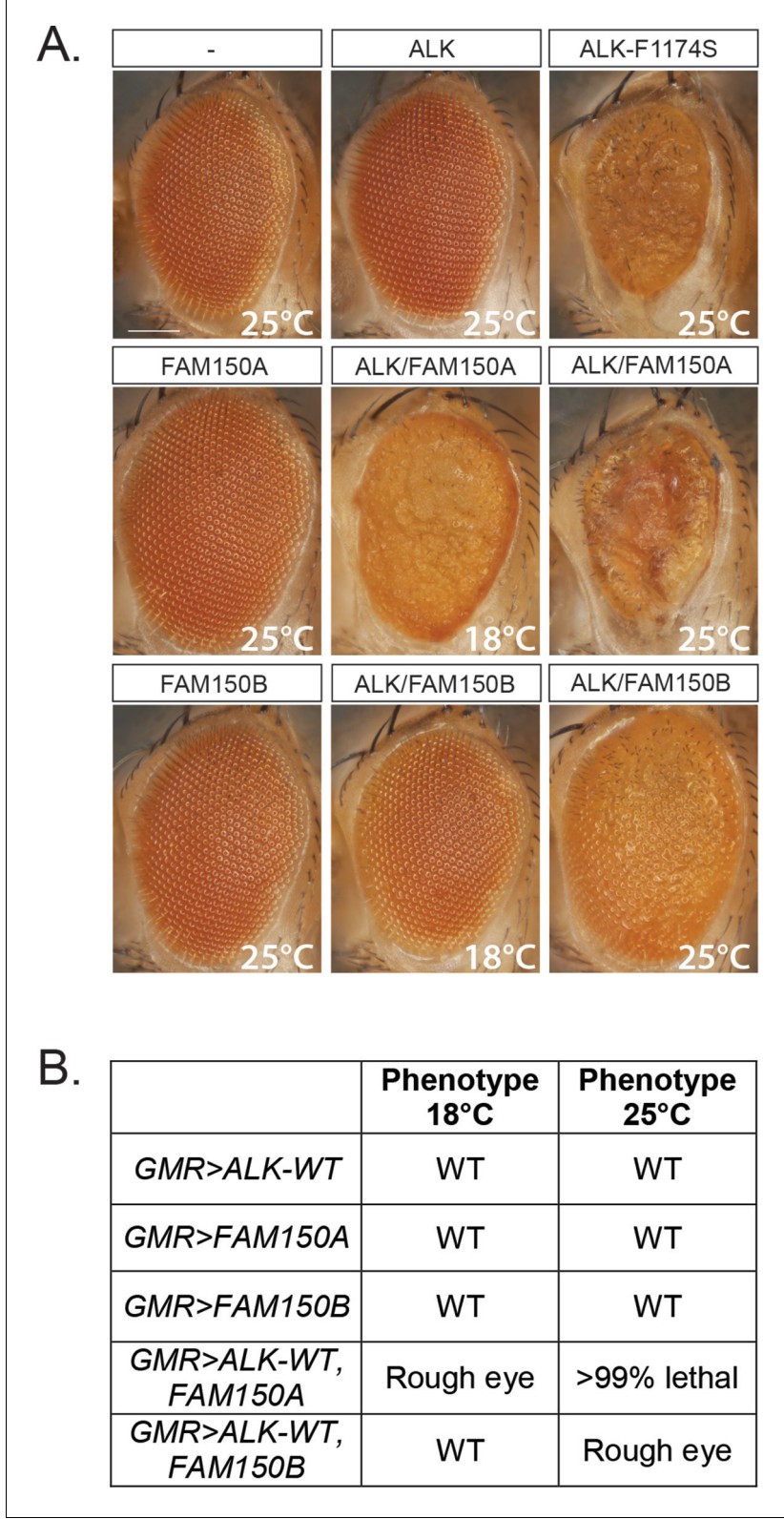

**Figure 3.** Expression of either FAM150A or FAM150B is sufficient for the activation of wild-type ALK. (**A**) Ectopic expression of either FAM150A or FAM150B together with the wild-type human anaplastic lymphoma kinase (ALK) receptor in the *Drosophila* eye with the *GMR-Gal4* driver disrupts the highly organized pattern of ommatidia of the *Drosophila* eye and generates a rough eye phenotype. Images of adult *Drosophila* eyes ectopically expressing

*Figure 3. Continued*

wild-type ALK in the presence of either FAM150A or FAM150B are shown. Controls expressing either wild-type ALK, FAM150A or FAM150B alone do not display a rough eye phenotype. The constitutively active ALK-F1174S mutant was employed as positive control. (**B**) Phenotypes observed in adult flies upon *GMR-Gal4* driven expression of either FAM150A or FAM150B together with the wild-type ALK at two different temperatures, 18°C and 25°C.

mAb135, *Figure 4—figure supplement 6*). These antibodies were next tested for their ability to modulate the activation of ALK by purified FAM150A. While mAb135 had no effect on FAM150A activation of ALK, we observed that mAb13 exhibited strong inhibition of FAM150A-induced ALK activation (*Figure 4E*), while mAb48 robustly activated ALK in the absence of FAM150A ligand (*Figure 4E*). Taken together, these data strongly support a role for the GR in the activation of ALK by FAM150 proteins.

In cell lines as well as in the *Drosophila* model, we observed extremely high levels of ALK activation when compared even with highly active ALK mutants such as ALK-F1174S, suggesting the possibility that these already active mutants can be activated to a higher level in the presence of a potent ligand. Therefore, we examined the ability of either FAM150A or FAM150B to 'superactivate' already active ALK mutants as described in neuroblastoma. In PC12 cells we indeed observed activation of the ALK-R1275Q mutant to a higher level, as evidenced by increased pERK1/2 and in neurite outgrowth assays (*Figure 4F*). A similar effect was observed with the ALK-F1174L mutant (*Figure 4—figure supplement 7*). This increased activity was also seen in CLB-GAR cells, which endogenously express the ALK-R1275Q mutant, with increased activation of ERK1/2 upon the addition of either FAM150A or FAM150B (*Figure 4G*).

In conclusion, our work identifies FAM150A and FAM150B as novel ligands that bind and activate the human ALK RTK. The identity of a ligand(s) for vertebrate ALK has been an evolving question for many years, with a number of reports over the past 15 years examining the role of the small heparin binding growth factors MDK and PTN as ligands for vertebrate ALK (*Stoica et al., 2001*; *Stoica et al., 2002*). However, a number of independent studies including our own work have presented convincing evidence that MDK and PTN do not activate ALK (*Motegi et al., 2004*; *Moog-Lutz et al., 2005*; *Mourali et al., 2006*; *Mathivet et al., 2007*; *Hugosson et al., 2014*; *Murray et al., 2015*). The recent identification of long chain heparins as activators of ALK in neuroblastoma cells (*Murray et al., 2015*) and the interplay of this with ALK activation by FAM150A and FAM150B will require further work. FAM150A and FAM150B are very basic proteins; FAM150A has a predicted pI of 10.6, while FAM150B has one of 9.75, and we show here that FAM150A exhibits heparin binding properties, however we have not observed a cooperative role of heparin and the FAM150 ligands in ALK activation as has been described for the fibroblast growth factor receptors and FGF. Our data also suggest that FAM150 ligands can bind an ALK-LTK heterodimer complex. While the physiological significance of such an interaction is currently unclear, ALK and LTK have been reported to be coexpressed in some tissues, such as the mouse hippocampus (*Weiss et al., 2012*), where such an interaction may be important.

FAM150 proteins do not seem to be conserved outside the vertebrates. In zebrafish there are three FAM150 proteins (FAM150ba, FAM150bb and FAM150A), while in mouse and humans only FAM150A and FAM150B exist (*Zhang et al., 2014*). In invertebrate model organisms the ALK ligands, Jeb in *Drosophila* and HEN-1 in *C. elegans*, do not appear to resemble the FAM150A/B ligands. The *Drosophila* Jeb ligand is unable to activate either mouse or human ALK (*Yang et al., 2007*), and to date, no Jeb-like ligand for vertebrate ALKs has been reported. The evolution of the FAM150 proteins as ALK ligands is an interesting topic for further investigation.

In the case of ALK mutations that are ligand dependent, as well as in situations where the ALK receptor is overexpressed, such as in neuroblastoma, the mechanisms by which ALK is activated is fundamental for understanding their role in tumor development. Here the identification of ALK ligands will allow a more rigorous interrogation of ALK signaling in these scenarios. The finding that FAM150A and FAM150B are not only able to activate the wild-type receptor, but also to 'super-activate' mutant ALK receptors, is important in this context. These findings, together with the observed expression pattern of FAM150A and FAM150B in adrenal gland and thyroid and the

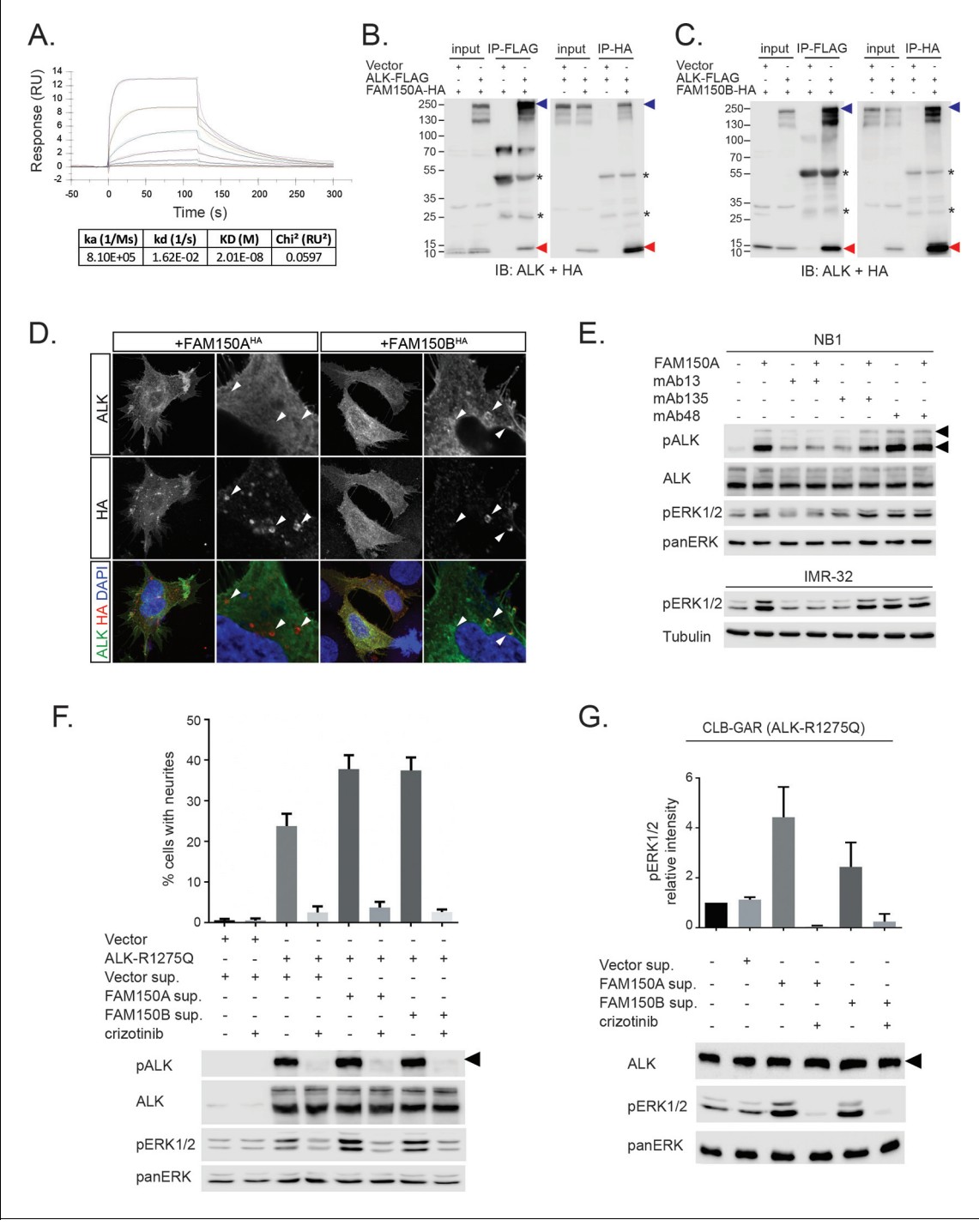

**Figure 4.** FAM150A and FAM150B bind to ALK and further activate signaling mediated by the R1275Q ALK neuroblastoma mutation. (**A**) Binding kinetics of purified FAM150A to extracellular domain of anaplastic lymphoma kinase (ALK-ECD-Fc) in a Biacore surface plasmon resonance (SPR) analysis. (**B**) FAM150A immunoprecipitates with human ALK. Immunoprecipitation with either anti-FLAG(DYKDDDDK)(ALK) or anti-HA (FAM150A) was performed and the resulting immunoprecipitates immunoblotted for the presence of ALK (blue arrowheads) and FAM150A (red arrowheads), *indicates immunoglobulin light and heavy chains. (**C**) FAM150B immunoprecipitates with human ALK. Immunoprecipitation with either anti-FLAG (ALK) or anti-HA (FAM150B) was performed and the resulting immunoprecipitates immunoblotted for the presence of ALK (blue arrowheads) and FAM150B (red arrowheads),*indicates immunoglobulin light and heavy chains. (**D**) Human Embryonic Kidney (HEK) 293 cells expressing ALK were incubated with either control or HA-tagged FAM150A or HA-tagged FAM150B conditioned

*Figure 4. continued on next page*

*Figure 4. Continued*

medium prior to analysis by immunohistochemistry. Both HA-tagged FAM150A and HA-tagged FAM150B bind to ALK-expressing cells. Higher magnification panels indicate intracellular vesicles positive for both ALK and HA-tagged FAM150A/B. (E) NB1 and IMR32 neuroblastoma cells were treated with 2 µg/ml monoclonal antibodies (mAB13, mAb48 or mAb135) prior to stimulation with FAM150A. Detection of ALK activation was visualized with pALK-Y1604 (arrowheads) and pERK1/2 in whole cell lysates. Pan-ERK or tubulin were employed for equal loading. (F) Whole cell lysates from PC12 cells expressing either vector control or ALK-R1275Q were stimulated with medium from HEK293 cells transfected with vector control, FAM150A or FAM150B prior to analysis by immunoblot. Analysis was carried out in the presence or absence of 250 nM crizotinib. Detection of ALK activation was visualized with pALK-Y1604 (arrowheads) and pERK1/2 in whole cell lysates. Pan-ERK was employed for equal loading. Neurite outgrowth was performed in triplicate and each sample within an experiment was performed in duplicate (error bars indicate SD). (G) CLB-GAR cells harboring the ALK-R1275Q mutant were stimulated for 30 min with medium from HEK293 cells transfected with either vector control, FAM150A or FAM150B prior to analysis by immunoblot. Analysis was carried out in the presence or absence of 250 nM crizotinib. Detection of ALK signaling activation was visualized with pERK1/2. Pan-ERK was employed for equal loading. pERK1/2 intensity was analyzed from three independent experiments (error bars indicate SD).

The following figure supplements are available for Figure 4:

**Figure supplement 1.** FAM150A binds to the extracellular domain of human ALK by ELISA.

**Figure supplement 2.** ALK interacts with both FAM150A and FAM150B.

**Figure supplement 3.** ALK and LTK interact in the presence of FAM150A. FAM150A and anaplastic lymphoma kinase (ALK) coimmunoprecipitate with leukocyte tyrosine kinase (LTK).

**Figure supplement 4.** Effect of deletion of the glycine rich domain of ALK on FAM150A and FAM150B binding

**Figure supplement 5.** Effect of glycine mutations in the glycine-rich domain of ALK on FAM150A and FAM150B binding.

**Figure supplement 6.** Identification of monoclonal antibodies recognising the glycine-rich region of the ALK ECD.

**Figure supplement 7.** FAM150A and FAM150B bind to ALK and further activate signaling mediated by the ALK-F1174L neuroblastoma mutant.

inhibition of ALK activation by FAM150A with monoclonal antibodies shown here, suggest that a future potential therapeutic arm may involve agents that interfere with FAM150A/B activation of ALK offering additional therapeutic approaches in patients with neuroblastoma for which single drug treatments targeting ALK have had limited effect to date.

## Materials and methods

### Antibodies and inhibitors

The primary antibodies used were anti-pan-ERK (1:10,000; BD Transduction Laboratories), anti-ALK (for immunofluorescence 1:1000; ab4061, Abcam), anti-ALK (D5F3, 1:5000; Cell Signaling Technology), anti-ALK mAb135 (1:2000; [*Witek et al., 2015*]), anti-pERK5 (1:1000; Cell Signaling Technology), anti-pALK-Y1278 (1:2000; Cell Signaling Technology), anti-pALK-Y1604 (1:2000; Cell Signaling Technology) and anti-pERK1/2-T202/Y204 (1:2000; Cell Signaling Technology), anti-FAM150A (1:4000, Atlas Antibodies), anti-HA (1:1000 for immunofluorescence, 1:6000 for immunoblotting; 16B12, Covance). The activating monoclonal antibody mAb46 and ALK monoclonal antibodies mAb13, mAb48 were a kind gift from M. Vigny and have been described previously (*Moog-Lutz et al., 2005*). The ALK inhibitor crizotinib was purchased from Chem Express (Shanghai).

## Immunofluorescence

Human embryonic kidney (HEK) 293 cells were grown on collagen-coated cover slips in 24-well plates, prior to Lipofectamine transfection with either pcDNA3-ALK or pcDNA3 alone as control. Conditioned medium containing FAM150A-HA or FAM150B-HA was then incubated for 20 min prior to fixation and immunostaining. For immunostaining of HEK cells, cells were fixed with 4% paraformaldehyde/Dulbecco's Modified Eagle Medium (DMEM) and blocked with 50 mM NH$_4$Cl/ phosphate buffered saline (PBS). After permeabilization with 0.3% Triton X-100 and 5% goat serum containing PBS, cells were incubated with primary antibody as indicated overnight at 4°C. For visualization cells were further incubated with fluorescence-labeled secondary antibody followed by analysis on LSM710 or LSM800 confocal microscopes (Zeiss).

## Cell lysis and immunoblotting

PC12 cells cotransfected with wild-type ALK, together with either FAM150A or FAM150B, were incubated for 24 hr prior to starvation in serum-free DMEM for a further 24 hr. PC12 cells expressing human wild-type ALK were serum starved for 24 hr prior to stimulation with FAM150A or FAM150B conditioned medium for 30 min. IMR-32 cells were stimulated with either 0.85 µg/ml mAb46 as positive control (*Moog-Lutz et al., 2005*), or with FAM150A or FAM150B conditioned medium for 20 min unless otherwise stated. For ALK inhibition, crizotinib was employed at a final concentration of 250 nM. Both PC12 and IMR-32 cells were lysed in 1 × sodium dodecyl sulfate (SDS) sample buffer, and precleared lysates were run on SDS-PAGE, followed by immunoblotting using the indicated antibodies. ALK downstream activation was detected by anti-pERK1/2. Pan-ERK was used as loading control. ALK phosphorylation was checked by using pALK-Y1278 and pALK-Y1604 antibodies. Cell lysis and immunoblotting was performed according to protocols described previously (*Schonherr, Yang et al., 2010*).

## Immunoprecipitation

To detect the interaction between ALK and FAM150A or FAM150B, HEK293 cells (in 10 cm dishes, 80–90% confluent) were transfected with 5 µg of pcDNA3-ALK-FLAG together with either 5 µg of pTT5-FAM150A-HA or pcDNA3-FAM150B-HA. As controls, cells were transfected with 5 µg of pcDNA3 vector together with 5 µg of pcDNA3-ALK-FLAG or pTT5-FAM150A-HA or pcDNA3-FAM150B-HA, correspondingly. Cells were lysed 16–20 hr after transfection, in 1 ml of lysis buffer (50 mM Tris-Cl, pH7.4, 250 mM NaCl, 1% Triton X-100 and proteinase inhibitor cocktail) on ice for 10 min prior to clarification by centrifugation at 14,000 rpm at 4°C for 15 min. Supernatant (30 µl) was taken and used as input control, and the remaining sample was incubated with either 20 µl of anti-FLAG M2 magnetic beads (Sigma) or 20 µl of anti-HA agarose beads (ThermoScientific) correspondingly for 2 hr at 4°C. The beads were then washed five times with lysis buffer and boiled in 80 µl of 1 x SDS sample buffer. Both inputs and immunoprecipitation products were separated by 9% SDS-PAGE gel. ALK and FAM150A or ALK and FAM150B were detected in the same blot with anti-ALK mAb135 (1:2000) and anti-HA antibody (1:6000, Covance), respectively.

## Neurite outgrowth assay

PC12 (2 x 10$^6$) cells were co-transfected with either 0.3 µg of empty pcDNA3 vector control, pcDNA3-ALK or pcDNA3-ALK-F1174L together with 0.5 µg pEGFPN1 (Clontech) and either 0.5 µg pTT5-FAM150A-HA or pcDNA3-FAM150B-HA as indicated by electroporation using Amaxa electroporator (Amaxa Biosystems). Cells were resuspended in 100 µl Ingenio electroporation solution (Mirus Bio LCC) prior to transfection. After transfection, cells were kept in minimum essential medium (MEM) supplemented with 7% horse serum and 3% fetal bovine serum and seeded into 24-well plates. After 24 hr of incubation, the fraction of Green fluorescent protein (GFP)-positive and neurite carrying cells versus GFP-positive cells was observed under a Zeiss Axiovert 40 CFL microscope. PC12 cells incubated with FAM150A-conditioned or FAM150B-conditioned media were analyzed for neurite outgrowth 24 hr after addition of conditioned media. For ALK inhibition, crizotinib was employed at a final concentration of 250 nM. To be judged as a neurite carrying cell, the neurite of the cell should be at least twice the diameter of a normal cell body. Experiments were performed in triplicates, and each sample within an experiment was performed in duplicate.

## Generation of FAM150A-conditioned and FAM150B-conditioned media

HEK293 cells, at ~90% confluency in 10 cm dishes, were transfected with 8.0 µg of either pcDNA3 vector control, pTT5-FAM150A-HA or pcDNA3-FAM150B-HA with Lipofectamine 2000 (Invitrogen). After 6 hr, medium was replaced with non-supplemented MEM and the subsequent conditioned medium was harvested 48 hr post transfection. For IMR-32 stimulation experiments 1 ml of either control, FAM150A-conditioned or FAM150B-conditioned medium was added to IMR-32 cells seeded in 6-well plates resulting in a total final volume of 3 ml. For PC12 stimulation experiments, medium was replaced with either control, FAM150A-conditioned or FAM150B-conditioned medium as indicated.

## Generation and purification of FAM150A-His

DNA encoding FAM150A (amino acids 21–129) was cloned into pFastBac-HBM TOPO (Invitrogen) and used to transform competent DH10Bac cells (Invitrogen) generating the recombinant FAM150A bacmid. Sf21 cells were transformed with the bacmid DNA using Cellfectin II reagent (Invitrogen) in serum-free Grace's Medium (Invitrogen). Supernatant from Sf21 cells containing recombinant FAM150A virus was collected after 72 hr and used for virus amplification. For protein expression, Sf21 cells were grown to a cell density of $1–1.5 \times 10^6$ cells/ml in Sf900-II medium (Gibco) containing 1% fetal bovine serum, penicillin and streptomycin, and infected with FAM150A recombinant viral stock. Medium was harvested 72 hr after infection and protein was affinity-purified on Ni-NTA agarose resin (Qiagen).

## Monoclonal antibody modulation of ALK activation by FAM150A

Wild-type ALK expressing neuroblastoma cell lines NB1 and IMR-32 were treated with either 2 µg/ml of FAM150A, 2 µg/ml of one of the following monoclonal antibodies: mAb13, mAb48, and mAb135, or combinations of FAM150A and either of the three antibodies respectively for 20 min. In case of the combination of FAM150A and antibodies, antibodies were added to the cells 30 min prior to addition of FAM150A. The activation of ALK by FAM150A was visualized with pALK-Y1604 and pERK1/2. Total ALK and tubulin were employed as internal controls.

## Expression of human ALK mutants in the *Drosophila* eye

The *Gal4-UAS* expression system was used for the ectopic expression of human ALK in *Drosophila* eye (*Brand and Perrimon 1993*). cDNAs encoding either *FAM150A* or *FAM150B* were cloned into *pUAST* and verified by sequencing. Transgenic flies carrying *pUAST-FAM150A* or *pUAST-FAM150B* were generated by BestGene, Inc., and crossed with *pGMR-Gal4, UAS-ALK-WT* (*Martinsson et al., 2011*), where hALK expression was driven by *pGMR-Gal4* (Bloomington Stock Center) in the developing eye. *UAS-ALK-F1174S* (*Martinsson et al., 2011*) driven by *pGMR-Gal4* was included as a positive control. Scale bar indicates 100 µm. Images were taken on a Zeiss AxioZoomV16 stereomicroscope.

## Binding of FAM150A to ALK-ECD-Fc by ELISA

FAM150A was purified as described (*Zhang et al., 2014*). For ELISA, 384-well white Maxisorp (Nunc Nalge) were coated overnight at 4°C with 20 µl of 2 µg/ml FAM150A diluted in 100 mM sodium bicarbonate buffer, pH 9.6. Subsequent steps were performed at room temperature. Plates were washed with PBST (PBS, 0.05% Tween20) and blocked for 1 hr with blocking/dilution buffer (PBST with 1% bovine serum albumin [BSA]). Plates were washed and 20 µl/well of ALK-Fc, LTK-Fc, and control-Fc diluted in blocking buffer added and incubated for 2 hr. Plates were washed and subsequently 20 µl/well of 1:10,000 dilution of horseradish peroxidase conjugated anti-human IgG (Jackson Labs) was added and incubated for 1 hr. After washing, 20 µl/well of ELISA Pico Chemiluminescent Substrate (Thermo Fisher Pierce) was added and incubated for 5 min before reading on a plate reader.

## Binding of FAM150A to ALK-ECD-Fc by Biacore analysis

Binding kinetics of FAM150A to LTK-ECD-Fc and ALK-ECD-Fc fusion proteins was determined using Biacore T100 SPR (GE Healthcare Life Sciences, Piscataway, NJ). LTK-ECD-Fc and ALK-ECD-Fc were captured on a CM4 sensor chip immobilized with anti-human IgG antibody using the human

antibody capture kit (GE Healthcare Life Sciences, Piscataway, NJ). 4-(2-hydroxyethyl)-1-piperazinee-thanesulfonic acid (HEPES) buffered saline10 mM, pH 7.4 with 0.05% Tween20 20 µg/ml heparin sodium salt (HBS-P, GE Healthcare Life Sciences, Piscataway, NJ, Sigma Aldrich, St Louis, MO) was used as the running and dilution buffer. Capture levels of the ECD-Fc fusions were adjusted to ~300–500 **resonance units** (RU). FAM150A was injected at eight concentrations (200, 66.6, 22.2, 7.4, 2.46, 0.82, 0.27 nM, and a 0 nM control) for 120 s and dissociation was followed for 180 s. The association constant, dissociation constant, and affinity for FAM150A for LTK and ALK Fc-ECD fusions were calculated using the Biacore T100 evaluation software package using standard double referencing technique and the 1:1 binding model.

## Materials and methods for figure supplements

### Heparin binding of purified FAM150A-His and FAM150A/B from conditioned media

Purified FAM150A-His (25 µg) in 20 mM Tris-HCl in a total volume of 1 ml was incubated with 20 µl of heparin-agarose (H0402, Sigma) overnight at 4°°C followed by 5 × washing with 20 mM Tris HCl, 50 mM NaCl. Bound proteins were eluted by boiling in 70 µl 20 mM Tris-HCl, 0.9 M NaCl with loading buffer and eluates were separated on 15% SDS-PAGE prior to immunoblot with anti-FAM150A antibody. Purified FAM150A-His 5 µg was employed as input control. Conditioned media from cells transfected with pcDNA3 control or pTT5-FAM150A-HA or pcDNA3-FAM150B-HA was incubated with 30 µl of heparin-agarose (H0402, Sigma) for 3 hr at 4°°C followed by 5 × washing with lysis buffer. Bound proteins were eluted by boiling in 50 µl 4× SDS sample buffer and eluates were separated on 13% SDS-PAGE prior to immunoblot with anti-HA antibody. Conditioned media 50 µl was employed as input control.

### Stimulation of ALK by FAM150A in the presence of heparin

IMR-32 cells were treated with either 1 µg/ml (final concentration) of purified FAM150A 753 alone or a combination of 1 µg/ml of FAM150A and 10 µg/ml of heparin (H3149, Sigma) 754 for the indicated time points (5, 10, 20, 30 and 60 min). Cell lysates were subjected to immunoblotting with anti-pERK1/2 antibody to visualize the activation effect. Tubulin was employed as loading control. NB1 cells were treated with heparin alone at the indicated concentrations (1, 10 and 100 µg/ml), 2 µg/ml of FAM150A alone, or a combination of 10 µg/ml of heparin and 2 µg/ml of FAM150A for 10 min prior to immunoblotting analysis. pALK (Y1604) and pERK1/2 antibodies were used to evaluate the activation effect. Total ALK and panERK were used as loading controls.

### Coimmunoprecipitation and immunoblotting

HEK293 cells (10 cm dishes, 80–90% confluent) were transfected with 5 µg of pcDNA3-ALK 5 µg of pTT5-FAM150A-HA, or 5 µg of pcDNA3-ALK 5 µg of pcDNA3-FAM150B-HA, respectively. Around 20 hr after transfection, cells were lysed, clarified and equally divided prior to incubation with either 2 µg of mouse IgG control (Santa Cruz), 2 µg of ALK mAb31, or 2 µg of mouse anti-HA (Covance) antibody. After overnight incubation at 4°°C, cell lysates with antibodies were further incubated with 15 µl of protein-G sepharose (GE healthcare) for another 2 hr. After 4 × washes with lysis buffer, agarose beads were boiled in 2 × SDS sample buffer. Eluates, as well as 1/50th of cell lysate (input), were separated on 9% SDS-PAGE. ALK and FAM150A/B were detected in the same blot employing anti-ALK mAb135 and anti-HA antibodies.

To investigate interaction between ALK, LTK and FAM150A, HEK293 cells in 10 cm dishes were transfected with 3 µg of pcDNA3-LTK-HA 4 µg of pTT5-FAM150A, or 3 µg of pcDNA3-LTK-HA 3 µg of pcDNA3-ALK 4 µg of pTT5-FAM150A, respectively. Around 20 hr after transfection, cells were lysed, clarified and equally divided prior to incubation with 30 µl of mouse IgG agarose (A0919, Sigma), or 15 µl of mouse anti-HA agarose (ThermoScientific), respectively, for 2 hr at 4°C. After 4 × washes, agarose beads were boiled in 2 × SDS sample buffer. Eluates, as well as 1/50th of cell lysate (input), were separated in 9% SDS-PAGE. ALK, LTK and FAM150A were detected in the same blot employing anti-ALK mAb135, anti-HA and anti-FAM150A antibodies.

To test whether the glycine-rich region (GR) of the ALK extracellular domain (ECD) is involved in binding of FAM150A/B, a GR-deleted version of ALK (ALK-delGR) was generated by overlap extension PCR to remove the entire GR of ALK (amino acids 689–940). In addition, ALK with mutations in

the conserved glycines G740D, G823D, G893E and G934D were also generated to test whether mutation of these conserved glycines affects the binding of FAM150A/B. For coimmunoprecipitation with ALK-delGR, HEK293 cells in 10 cm dishes were transfected with 4 µg of pcDNA3-ALK 4 µg of pTT5-FAM150A-HA, or 4 µg of pcDNA3-ALK-delGR 4 µg of pTT5-FAM150A-HA, or 4 µg of pcDNA3-ALK 4 µg of pcDNA3-FAM150B-HA, or 4 µg of pcDNA3-ALK-delGR 4 µg of pcDNA3-FAM150B-HA, respectively. For coimmunoprecipitation with different ALK glycine mutants, 4 µg of pTT5-FAM150A-HA or 4 µg of pcDNA3-FAM150B-HA were cotransfected with 4 µg of pcDNA3-ALK (WT) or ALK bearing different glycine mutations (G740D, G823D, G893E and G934D) into HEK293 cells in 10 cm dishes, respectively. The same coimmunoprecipitation procedure was followed as described above for ALK, LTK and FAM150A. ALK and FAM150A/B were detected in the same blot employing anti-ALK (D5F3) and anti-HA antibodies.

## Characterization of monoclonal antibodies recognizing the ALK ECD

HEK293 cells transfected with control vector, ALK and ALK-delGR were lysed 24 hr after transfection. Cell lysates were subjected to immunoblotting analysis with different ALK monoclonal antibodies (mAb13, mAb48 and mAb135 recognizing the extracellular portion of ALK; D5F3 recognizing the intracellular portion of ALK).

## Acknowledgements

The authors acknowledge the important contribution to this study, particularly in the spirit of free exchange of scientific reagents and knowledge, made by Lewis T. Williams who generously provided reagents and facilitated collaboration in this study. We thank the Centre for Cellular Imaging (CCI) at the University of Gothenburg for providing confocal imaging support and Anne Uv for critical reading of the manuscript. This work was supported by grants from the Swedish Cancer Society (BH 12-0722, RHP 12-0796), the Children's Cancer Foundation (BH 14/150, RHP 13/0049), the Swedish Research Council (RHP 621-2011-5181, BH 521-2012-2831), and the SSF Programme Grant (RB13-0204).

## Additional information

### Funding

| Funder | Grant reference number | Author |
| --- | --- | --- |
| Cancerfonden (Swedish Cancer Society) | 12-0796 | Ruth H Palmer |
| Vetenskapsrådet | 621-2011-5181 | Ruth H Palmer |
| Barncancerfonden (Swedish Childhood Cancer Foundation) | 13/0049 | Ruth H Palmer |
| Stiftelsen för Strategisk Forskning | 13-0204 | Bengt Hallberg Ruth H Palmer |
| Cancerfonden (Swedish Cancer Society) | 12-0722 | Bengt Hallberg |
| Vetenskapsrådet | 521-2012-2831 | Bengt Hallberg |
| Barncancerfonden (Swedish Childhood Cancer Foundation) | 14/150 | Bengt Hallberg |

The funders had no role in study design, data collection and interpretation, or the decision to submit the work for publication.

### Author contributions

JG, GU, YY, GW, PM, KP, FH, HZ, AWH, RH, BH, RHP, Conception and design, Acquisition of data, Analysis and interpretation of data, Drafting or revising the article; AM, Acquisition of data, Analysis and interpretation of data

### Author ORCIDs

Ruth H Palmer, http://orcid.org/0000-0002-2735-8470

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
