## [Decision Letter]

Thank you for submitting your work entitled "FAM150A and FAM150B are activating ligands for Anaplastic Lymphoma Kinase" for peer review at *eLife*. Your submission has been favorably evaluated by Tony Hunter (Senior Editor) and three reviewers, one of whom is a member of our Board of Reviewing Editors.

The reviewers have discussed the reviews with one another and the Reviewing Editor has drafted this decision to help you prepare a revised submission.

The authors present evidence that FAM150A and FAM150B are activating ligands for ALK. The authors previously showed that FAM150A and FAM150B bind LTK. Based on similarity between LTK and ALK in the membrane proximal domain, the authors hypothesized that FAM150A and FAM150B also bind ALK. They demonstrate that conditioned medium containing FAM150A and FAM150B can activate ALK, that FAM150A and ALK co-IP when overexpressed, that expression of FAM150A and FAM150B and ALK in fly eyes induces a phenotype consistent with an interaction and biophysical studies that suggest that FAM150A and ALK bind.

The identification of ligands for the ALK RTK is important, particularly because of the involvement of ALK mutations in human cancer. The authors present some evidence that is consistent with the conclusion that FAM150A and FAM150B act as ALK ligands. However, the presented evidence falls short of definitively supporting this conclusion.

1) The structure of the paper should be revised. There is no Introduction and the first paragraph of the Results section represents an extended abstract. Moreover, the findings should be presented in the past tense.

2) The authors' premise for these studies is that the membrane proximal Gly-rich extracellular region of ALK represents the FAM150A/B binding site based on homology with LTK. This should be tested directly.

3) Do mutations of conserved glycines (Figure 1—figure supplement 1) reduce FAM150A/B binding?

4) Do available agonist/antagonist ALK antibodies interfere with FAM150A/B binding?

5) The authors state that heparin was recently described as an activating ligand for ALK and speculate that heparin and FAM150 may act synergistically. This possibility should be tested experimentally using purified FAM150.

6) The authors determined the Kd for binding of FAM150A to ALK to be approximately 20 nM, which is a rather low affinity compared to most other RTK-ligand interactions. Moreover, from Figure 3—figure supplement 1, it appears as if the affinity of FAM150A is about 50-fold higher for binding to LTK than for binding to ALK. Is the rather low affinity interaction between FAM150 and ALK of physiological importance? Is there a co-factor involved (see point 5 above)?

7) Can endogenous interactions between FAM150A/B and ALK be demonstrated? If not, this should be noted as a deficiency of the study.

---

## [Author Response]

The identification of ligands for the ALK RTK is important, particularly because of the involvement of ALK mutations in human cancer. The authors present some evidence that is consistent with the conclusion that FAM150A and FAM150B act as ALK ligands. However, the presented evidence falls short of definitively supporting this conclusion.

1) The structure of the paper should be revised. There is no Introduction and the first paragraph of the Results section represents an extended abstract. Moreover, the findings should be presented in the past tense. 

The structure of the paper has now been amended as suggested by the reviewers.

2) The authors' premise for these studies is that the membrane proximal Gly-rich extracellular region of ALK represents the FAM150A/B binding site based on homology with LTK. This should be tested directly. 

As suggested, the importance of the glycine-rich region of ALK in the interaction with FAM150A/B has been examined. Although we are concerned that deletion of such a large portion of the ALK ECD will have serious repercussions for the receptor we have tested the ability of either FAM150A or FAM150B to bind to a glycine-rich region deletion mutant of ALK (ALK-DelGR). We were surprised to find that the binding of FAM150B to ALK-DelGR was severely impaired, while FAM150A is still able to bind (Figure 4—figure supplement 4).

Further experiments, outlined in more detail in points 3 and 4 below, strongly support an important role of the glycine-rich domain in the binding and activation of ALK by the FAM150 proteins. Firstly (related to point 3 below), mutation of individual glycines in the ALK ECD abolishes binding FAM150B to ALK (Figure 4—figure supplement 5). Once again, as we would predict from the analysis of the ALK glycine-rich deletion mutant, interaction of FAM150A with ALK single glycine mutations is still observed. Secondly (related to point 4 below), we were able to identify antibodies recognizing the ALK glycine-rich domain which block activation of ALK by FAM150A (Figure 4 and Figure 4—figure supplement 6).

3) Do mutations of conserved glycines (Figure 1—figure supplement 1) reduce FAM150A/B binding? 

To perform this experiment we mutated each of the four conserved glycines in the human ALK ECD that are known to result in a loss of function ALK receptor in *Drosophila* (Lorén et al., 2003). In the human ALK receptor these residues are G740, G823, G893 and G934. We show that while FAM150A binding to ALK is still observed in the glycine mutants as well as the WT ALK receptor, the binding of FAM150B is abolished (Figure 4—figure supplement 5). Currently we do not understand the molecular mechanisms underlying the discrepancy between FAM150A and FAM150B binding that we observe; however these data suggest that there are differences in FAM150A and FAM150B interactions with the ALK.

4) Do available agonist/antagonist ALK antibodies interfere with FAM150A/B binding? 

We have investigated this in the revised manuscript. We have a number of antibodies generated towards the ALK ECD (Moog-Lutz, JBC, 2005, Witek, PLoS ONE, 2015) which we have tested. Initially we examined which of these were able to recognize the ALK glycine-rich region (Figure 4—figure supplement 6). These were then further tested for their ability to interfere with the activation of ALK by FAM150A/B. We were able to identify one anti-ALK mAb (mAb13) that bound to the glycine-rich region of ALK and effectively blocks activation of endogenous ALK by FAM150A (Figure 4). This was not exclusive to one cell neuroblastoma cell line, as we could show this inhibition of ALK activation by FAM150A in both IMR-32 and NB1 cells.

5) The authors state that heparin was recently described as an activating ligand for ALK and speculate that heparin and FAM150 may act synergistically. This possibility should be tested experimentally using purified FAM150. 

We have investigated the activation of ALK by heparin, in the presence and absence of either FAM150A/B. We have examined this in PC12 cells transfected with ALK as well as in NB1 and IMR-32 neuroblastoma cell lines that express WT ALK. We have been unable to show either activation of ALK by heparin alone, nor any synergistic effect of heparin together with the FAM150A/B ligands. We have tried this in cells with (i) FAM150A/B conditioned medium, (ii) purified FAM150A proteins, and additional with experiments in which cells were plated on heparin coated plates. In all cases we saw a slight inhibition of FAM150A/B activation upon addition of heparin, rather than a synergistic activation. We also examined the effect of heparin in ELISA, where there was comparable binding of FAM150A with the addition of heparin (data not shown). Some of this data is shown in Figure 2—figure supplement 2. Thus, at the present time we are unable to show activation of WT ALK with heparin, however, we have observed a strong interaction between the FAM150 proteins and heparin, with both FAM150A and FAM150B binding strongly to heparin-agarose (Figure 2—figure supplement 1). Since the FAM150A and FAM150B proteins are very basic in nature, with a predicted pI of 10.6 for FAM150A and 9.75 for FAM150B), we suspect that heparin, or heparin modified proteoglycans may be involved in the activation of ALK by FAM150 proteins. This will clearly be an important issue to clarify in the future.

6) The authors determined the Kd for binding of FAM150A to ALK to be approximately 20 nM, which is a rather low affinity compared to most other RTK-ligand interactions. Moreover, from Figure 3—figure supplement 1, it appears as if the affinity of FAM150A is about 50-fold higher for binding to LTK than for binding to ALK. Is the rather low affinity interaction between FAM150 and ALK of physiological importance? Is there a co-factor involved (see point 5 above)? 

A range of ligand affinities for different RTKs and their ligands have been reported, often higher that the 20nM we report here for ALK and FAM150A. One of the most studied ligand-RTK receptor pair examples is EGF-EGFR, where one published estimate is a binding affinity of 6nM (Ozcan et al., PNAS, 2006). The reviewer is correct that the affinity of ALK for the FAM150A proteins is significantly lower than for LTK. In our experiments we consistently note that FAM150B is a very strong activator and binder of ALK, and suspect that FAM150B may bind ALK very strongly. Unfortunately we have been unable to produce purified FAM150B protein and thus have not been able to directly measure affinity of the FAM150B-ALK interaction. This is something that we hope to be able to address in the future, but are currently unable to do so due to technical difficulties.

Based on these results, and the comment from the reviewer, we examined the interplay between FAM150A, ALK and LTK. We show that both FAM150A and ALK are immunoprecipitated with LTK (Figure 4—figure supplement 3). Thus, we suggest in the revised version of the manuscript that ALK and LTK are potentially able to form a complex binding to FAM150A. Indeed, these proteins are known to be co-expressed in some neural tissues, such as the mouse hippocampus (Weiss et al., Pharmacol, Biochem Behav, 2011), and such an interaction may be of important in this context. This observation will require further studies to understand the physiological significance of such an interaction.

7) Can endogenous interactions between FAM150A/B and ALK be demonstrated? If not, this should be noted as a deficiency of the study. 

Despite extensive efforts we are unable to show endogenous interaction between FAM150A/B. We are hesitant to agree that this should be noted as a deficiency of the study since very few studies show interaction of a secreted peptide ligand for a receptor tyrosine kinase through immunoprecipitation with its ligand at the level of endogenous proteins. Furthermore, while we have excellent ALK antibodies, we have no antibody that recognizes FAM150B, and our FAM150A antibodies work only on immunoblotting. Thus, while we have attempted to show endogenous interaction, we do not currently have the optimal tools to test this robustly.